# Immunoglobulin A as a Key Immunological Molecular Signature of Post-COVID-19 Conditions

**DOI:** 10.3390/v15071545

**Published:** 2023-07-13

**Authors:** Graziele F. Sousa, Raphael M. Carpes, Carina A. O. Silva, Marcela E. P. Pereira, Amanda C. V. F. Silva, Viktoria A. G. S. Coelho, Evenilton P. Costa, Flávia B. Mury, Raquel S. Gestinari, Jackson Souza-Menezes, Manuela Leal-da-Silva, José L. Nepomuceno-Silva, Amilcar Tanuri, Orlando C. Ferreira-Júnior, Cintia Monteiro-de-Barros

**Affiliations:** 1Laboratório de Campanha para Testagem e Pesquisa do COVID-19 (LCC), Instituto de Biodiversidade e Sustentabilidade (NUPEM), Universidade Federal do Rio de Janeiro (UFRJ), Macaé 27965-045, RJ, Brazil; 2Laboratório de Virologia Molecular, Departamento de Genética, Instituto de Biologia, Universidade Federal do Rio de Janeiro (UFRJ), Rio de Janeiro 21941-902, RJ, Brazil; 3Cintia Monteiro de Barros, Instituto de Biodiversidade e Sustentabilidade (NUPEM), Universidade Federal do Rio de Janeiro (UFRJ), Av. São José do Barreto 764, Macaé 27965-045, RJ, Brazil

**Keywords:** post-COVID-19 conditions, immunoglobulin profiles, SARS-CoV-2

## Abstract

COVID-19 has infected humans worldwide, causing millions of deaths or prolonged symptoms in survivors. The transient or persistent symptoms after SARS-CoV-2 infection have been defined as post-COVID-19 conditions (PCC). We conducted a study of 151 Brazilian PCC patients to analyze symptoms and immunoglobulin profiles, taking into account sex, vaccination, hospitalization, and age. Fatigue and myalgia were the most common symptoms, and lack of vaccination, hospitalization, and neuropsychiatric and metabolic comorbidities were relevant to the development of PCC. Analysis of serological immunoglobulins showed that IgA was higher in PCC patients, especially in the adult and elderly groups. Also, non-hospitalized and hospitalized PCC patients produced high and similar levels of IgA. Our results indicated that the detection of IgA antibodies against SARS-CoV-2 during the course of the disease could be associated with the development of PCC and may be an immunological signature to predict prolonged symptoms in COVID-19 patients.

## 1. Introduction

COVID-19 is a disease that has infected millions of people worldwide and became a pandemic at the end of 2019 [1]. As of March 2023, more than six million people have died across the world, and many survivors have developed sequelae [2,3]. Most of the sequelae belong to a persistent group of symptoms that begin in the acute phase of infection with SARS-CoV-2. On the other hand, there is a group of symptoms that begin after the acute phase of COVID-19. These persistent conditions and incomplete recovery from COVID-19 are impacting healthcare systems worldwide, increasing the emotional, physical, and financial burden on patients, caregivers, and society [4]. These conditions are commonly referred to as post-COVID-19 condition (PCC), post-acute COVID-19 syndrome, post-acute sequelae of COVID-19 (PASC), or long COVID [5]. The World Health Organization (WHO) defines PCC as a condition characterized by symptoms that interfere with the patient’s daily life and that occurs after a history of probable or confirmed SARS-CoV-2 infection. However, the causes and mechanisms of PCC are under investigation and some groups correlate persistence of symptoms with detectable viral RNA levels. Based on the literature, PCC can be further divided into two categories. First, subacute COVID-19, which includes symptoms that occur 4–12 weeks after the end of acute COVID-19, and second, chronic post-COVID-19, which includes symptoms that occur 12 weeks after acute COVID-19 and cannot be attributed to other diseases [6].

PCC is associated with a wide range of symptoms. Studies in Europe, the United States, and China have found that approximately 87.4% of patients reported persistent symptoms after hospitalization, including fatigue, dyspnea, joint pain, chest pain, and psychological distress such as anxiety, depression, and mental concentration difficulties [7,8,9,10]. This wide range of symptoms related to PCC involves multiple organs and systems and involves factors such as hospitalization, sex, comorbidity, vaccination, and age, among others [11,12,13,14,15]. Therefore, to understand the heterogeneity of PCC, it is crucial to understand its etiology, risk factors, and immunological predictors. Previous studies suggest that the immunological system may be key to understanding the severity of the disease as well as the risk of developing PCC [4]. Previous work has described changes in the profile of T cells, including exhausted T cells, reduced numbers of CD4+ and CD8+ effector memory cells, high levels of autoantibodies, insufficient production of neutralizing IgG antibodies, and increased production of IgA [16,17,18]. The objective of this work was to describe the main symptoms developed by PCC patients and their serological characteristics (i.e., anti-SARS-CoV-2 antibodies) and to identify some signatures (molecular or conditions) of the development of PCC.

## 2. Materials and Methods

### 2.1. Experimental Ethical Criteria

Patients who sought care at the Centro de Acolhimento e Reabilitação Pós-COVID-19 (CARP) in the city of Macaé, in Rio de Janeiro, Brazil, were invited to participate in the study. A total of 252 patients sought care, and 151 agreed to sign the Informed Consent Form (ICF) and participate in the study. This work was developed with the approval of the Research Ethics Committee of the Universidade Federal do Rio de Janeiro campus Macaé (CEP/UFRJ-Macaé; CAAE protocol: 57373422.8.0000.5699). The inclusion criteria for volunteers were a positive diagnosis of COVID-19 and signed informed consent. The exclusion criteria were children and volunteers under 18 years old and pregnant women.

### 2.2. Patient Screening and Trial Groups

Post-COVID-19 volunteers were recruited from January 2022 to February 2023 and underwent anamneses by a physician at CARP and answered a questionnaire. Subsequently, volunteers were categorized into acute (0–4 weeks), subacute (5–12 weeks), or chronic (>12 weeks) phases [6] according to the type and duration of symptoms after the onset of COVID-19. For the serological analysis, blood samples were collected and divided into four groups: two different control groups and two experimental groups. The control groups were classified according to the presence or absence of COVID-19 infection: SARS-CoV-2 unexposed control (which included blood samples collected in January 2020 from the blood bank of the Municipal Hemotherapy Service before the first case of COVID-19 in Brazil) and nPCC volunteers, corresponding to individuals previously infected with SARS-CoV-2, without clinical signs of PCC symptoms. The experimental groups were classified according to hospitalization during the course of SARS-CoV-2 infection and the phase of PCC (subacute or chronic): PCCnh and PCCh subjects (Figure 1). We consider hospitalization to be a measure of COVID-19 severity, ranging from cases where the patient was under observation in the wards for one day to the most severe cases where the patient was hospitalized for several days with mechanical ventilation and intubation.

### 2.3. Blood Sample Processing and Storage

Blood samples were collected by peripheral venipuncture into 4 mL EDTA K3 tubes (Firstlab), and plasma was isolated by centrifugation at 3000× *g* (Daiki DT-5000) for 10 min and stored at −20 °C until further analysis.

### 2.4. S-UFRJ ELISA for the Detection of Anti-S Immunoglobulins

The S-UFRJ ELISA was performed using the trimeric spike glycoprotein of SARS-CoV-2 produced in stable transfected HEK293 cells, as described by Alvim et al. (2022) [19] and provided by the Cell Culture Engineering Laboratory (LECC) of the Federal University of Rio de Janeiro (UFRJ). The secondary antibody concentration used was 1:200 goat anti-human IgM (Fc) HRP-conjugated antibody (#MFCD00162459; Sigma, St. Louis, MO, USA), anti-human IgG (Fc) HRP-conjugated antibody (#SAB3701282; Sigma, St. Louis, MO, USA), and goat anti-human IgA HRP-conjugated antibody (#A18781; Invitrogen, Waltham, MA, USA) incubated for 1 h at room temperature. The reaction was read spectrophotometrically at 450 nm in a Multiskan Sky microplate reader (Thermo Scientific). The results were expressed either as optical density units (OD) or as the ratio of sample OD/cut-off value. The pre-pandemic healthy volunteers (blood samples collected in January 2020) were selected as negative controls. After optimization by means of a receiver operator characteristic (ROC) analysis, the cut-off was defined as the sum of the OD mean of negative controls in the same plate plus 3 times the OD standard deviation determined when 90 negative controls were tested [20].

### 2.5. Statistical Analysis

Data analysis was performed using GraphPad Prism 8 software (GraphPad Software Inc., Boston, MA, USA), and graphs were expressed as mean ± standard error of the mean, frequencies, and percentages. Gaussian distribution was verified using the Kolmogorov–Smirnov test. The Mann–Whitney U test was then used to compare two groups, and the Kruskal–Wallis test was used to compare more than two groups, followed by Dunn’s post hoc test. Differences were considered significant when *p* < 0.05. Proportional hazard regression models were used to obtain odds ratios (ORs) for each factor analyzed (comorbidities, vaccination, and hospitalization) among the PCC volunteers with an outcome of symptoms over 12 weeks after SARS-CoV-2 infection. The co-occurrence of symptoms was defined by Jaccard’s coefficient ranging from 0 to 1. Values closer to 1 indicate higher similarity indices. The Section 2 should provide sufficient information to allow replication of the results. Begin with the Section 2.1 describing the objectives and design of the study, as well as the pre-specified components.

## 3. Results

### 3.1. Baseline Descriptions

A total of 255 patients sought medical attention at the CARP, but only 151 agreed to participate and signed the ICF. All volunteers had a previous COVID-19 infection confirmed by RT-qPCR. From this total number of volunteers, the participation of women was almost three times higher than that of men (74.17% vs. 25.83%; Appendix A). The mean age of the volunteers was 52 years old (18–80 years), whereas the mean age of the chronic group was older than that of the subacute group (52.19 ± 13.86 vs. 49.81 ± 16.90 years; Appendix A). The mean duration of PCC was 341.1 days (32–1029 days), and more women (*n* = 109 of 151; 72.19%) sought medical assistance at CARP compared to men (*n* = 42 of 151; 27.81%). The baseline data, including age, sex, immunoglobulin production symptoms, and comorbidities domains, are shown in Appendix A.

### 3.2. Post-COVID-19 Conditions and Symptoms

The symptoms presented by the volunteers are identified in Figure 2. A total of 30 symptoms were described and further categorized into acute, subacute, and chronic according to the period of occurrence (Figure 2A,B). The most common clinical manifestations registered during acute illness were myalgia (62.25%), fatigue (61.59%), headache (60.26%), anosmia (60.26%), and ageusia (59.60%; Figure 2B). The most common symptoms reported by the PCC patients were fatigue (80.13%), myalgia (80.13%), memory loss (75.50%), hypoprosexia (72.19%), headache (72.19%), anxiety (68.21%), ageusia (64.90%), anosmia (63.58%), and difficulty reasoning (60.26%; Figure 1A). In the subacute phase of PCC, the most common symptoms were myalgia (58.94%), fatigue (56.29%), hypoprosexia (56.29%), memory loss (54.97%), and anxiety (52.98%). Taking into account the frequency of the most common symptoms (>50%), they were, on average, 1.4 times more frequent in the subacute phase compared to the chronic phase, with the exception of memory loss and hypoprosexia (Figure 2B). Interestingly, memory loss, hypoprosexia, and anxiety increased in frequency when comparing the acute, subacute, and chronic phases, being more prevalent in the last phase, in which myalgia and fatigue persisted in more than 80.1% of the PCC patients (Figure 2B).

Overall, the vaccination status of the volunteers was analyzed considering the COVID-19 diagnosis. At the time of infection, 41.67% of volunteers were unvaccinated, 6.94% had received at least one dose, 30.56% had received at least two doses, and 20.83% had received three doses. Analyzing the vaccinated volunteers, the difference between the date of the last vaccination and the date of infection was 5 ± 0.037 months. The period between vaccination (first, second, or third dose) and the interview ranged from 0 to 334 days and 0 to 420 days in the subacute and chronic phases, respectively. Figure 2C shows that unvaccinated patients were more frequent in the chronic group (71.55%) than in the subacute group (24%); therefore, the lowest frequency of symptoms was observed in the patients who received more doses of vaccine.

Based on the symptoms and PCC phases, we evaluated the antecedent factors (hospitalization, vaccination, and comorbidities) by odds ratios (ORs). The graph shows that unvaccinated individuals had higher odds of developing chronic PCC (OR 6.33 (95% CI 2.35–17.03)), followed by the severity of illness factor (hospitalization) (OR 2.80 (95% CI 0.78–9.97)) and comorbidities (OR 0.65 (95% CI 0.23–1.88)), which had the lowest magnitude (Figure 2D).

Therefore, a co-occurrence analysis of symptoms was performed using similarity indices and pairs of symptoms. The highest co-occurrences were concentrated among the nine most common symptoms reported by PCC patients. Overall, ageusia and anosmia had the highest co-occurrence (0.865), followed by hypoprosexia and memory loss (0.851). Symptoms with lower frequencies showed a clear co-occurrence, such as sleep disorder and weight change (0.546), and finally, angina and arrhythmia (0.509; Figure 3). In addition, the co-occurrence of symptoms based on sex showed subtle differences. The female group showed a high co-occurrence of memory loss and hypoprosexia (0.883) and memory loss and reasoning difficulties (0.794), compared to men (0.741 and 0.577, respectively). Anxiety and myalgia were the most common symptoms in women, showing co-occurrence with almost all symptoms related to them. The male group showed a high co-occurrence of anosmia and ageusia (0.955) compared to women (0.842). Finally, alopecia and depression were symptoms with a lower co-occurrence in the male group (Appendix A). The co-occurrence of symptoms taking account of the phases of PCC, revealed that aphasia and hoarseness in the subacute phase and anosmia and ageusia in the chronic phase had the highest co-occurrence (1.00 and 0.886, respectively). The chronic phase showed higher Jaccard index values for the nine most frequent symptoms compared to the subacute phase (Appendix A).

### 3.3. Post-COVID-19 Conditions and Comorbidities

Although no association was found between comorbidities and the development of PCC, we categorized the reported comorbidities based on the literature [21] and evaluated the frequency and the risk factor of comorbidity groups (Figure 4A). The most common group of comorbidities reported by patients were cardiovascular (*n* = 89 of 151; 58.94%) and metabolic (*n* = 50 of 151; 33.11%). Hypertension (47.02%) and diabetes (29.80%) were the most common comorbidities observed in the cardiovascular and metabolic groups, respectively (Figure 4A,B). When evaluating the risk factor of chronic PCC in relation to comorbidity groups, we observed that the metabolic (OR 1.14 (95% CI 0.46–2.85)) and neuropsychiatric (OR 1.29 (95% CI 0.40–4.12)) groups presented high chances of developing PCC (Figure 4C). Considering the high frequency of hypertension and diabetes in the PCC patients, we performed an OR analysis with these symptoms (combined and separated). Thus, diabetes was evaluated as a risk factor for the development of chronic PCC (OR 1.22 (95% CI 0.47–3.16); Appendix A).

### 3.4. Post-COVID-19 Conditions and Immunoglobulin Profile

The anti-S glycoprotein SARS-CoV-2 IgM, IgG, and IgA profiles were analyzed in the PCC patients to better understand the host’s prolonged serological response to SARS-CoV-2. Thus, an initial semi-quantitative analysis of IgM, IgA, and IgG was performed, taking into account the presence or absence of comorbidities, as shown in Figure 3D–F. The samples with optical density (OD) values below the cut-off were considered as not producing immunoglobulins. While all the patients were IgG positive for SARS-CoV-2, only a portion of them were IgA positive, in contrast to IgM, for which all the patients were negative, indicating that they were not in the acute phase of COVID-19 and were post-COVID-19 patients. The mean plasma levels (OD) of IgM, IgA, and IgG were 0.132 ± 0.005 OD, 0.735 ± 0.032 OD, and 1.975 ± 0.038 OD, respectively, in the patients with comorbidities. In the absence of comorbidities, these values were 0.129 ± 0.010 OD, 0.584 ± 0.053 OD, and 2.005 ± 0.054 OD, respectively. IgM and IgG levels did not show significant differences in relation to the presence or absence of comorbidities (Figure 4D,F). However, IgA levels were higher in patients with comorbidities compared to those without (Figure 4E). An analysis was performed to evaluate the correlation between comorbidity domains or quantity of symptoms and production of IgA; however, no correlation was found (Appendix A).

Second, the IgM, IgA, and IgG concentrations were evaluated in samples from patients with PCC and compared to the non-PCC (nPCC) control group. The production of IgM in PCC patients (0.131 ± 0.005 OD) was below the cut-off values, while IgG in the same group (1.982 ± 0.031 OD) was above the cut-off values for all samples analyzed, and none of the two immunoglobulins showed significant differences between the nPCC and PCC groups (Figure 5A,C). IgA levels showed a significant difference between the nPCC and PCC groups, indicating an increase in the plasma levels of SARS-CoV-2 IgA in PCC patients (0.639 ± 0.028 OD) (Figure 5B).

Third, the IgM, IgA, and IgG production in the PCC group was evaluated taking account of sex and did not show any statistical differences between men and women (Figure 5D–F). The IgM and IgG production was also analyzed considering three age groups and showed no significant differences (Figure 5G,I). Interestingly, only IgA production showed significant differences when comparing the three age groups (18–39, 40–50, and ≥60 years old). The highest IgA production was observed in the oldest group, and a significant difference was observed for the 40–59 (0.664 ± 0.040 OD) and ≥60 (0.671 ± 0.048 OD) age groups compared to the youngest group (0.431 ± 0.046 OD; Figure 5H).

Previous studies have reported that the severity of COVID-19 could be measured by hospitalization, and here we compared immunoglobulin production considering hospitalization. Anti-SARS-CoV-2 IgM levels were not significantly different in PCC patients compared with nPCC subjects, nor did they show a different trend from that observed when considering severity of illness (hospitalization) (Figure 6A). Analysis of anti-SARS-CoV-2 immunoglobulin production in PCC patients according to the occurrence of hospitalization revealed that IgG production only increased in the group of hospitalized subjects with PCC (PCCh) (2.186 ± 0.046 OD) compared to the nPCC group (1.972 ± 0.039 OD; Figure 6C). Interestingly, IgA levels increased in the group of non-hospitalized subjects with PCC (PCCnh) (0.588 ± 0.030 OD) compared to the nPCC group (0.374 ± 0.030 OD). The same occurred when comparing IgA in the PCCh (0.566 ± 0.047 OD) and nPCC groups (Figure 6B).

We then evaluated the production of anti-SARS-CoV-2 immunoglobulins in PCC patients divided into subacute and chronic phases, as well as with and without hospitalization. Similar to the previous data, IgM was not significantly different in the groups analyzed (Figure 6D). Regarding IgG production, PCCnh patients in the subacute phase showed higher production compared to nPCC subjects in the subacute phase (2.049 ± 0.435 OD). The nPCC group showed a significant difference between the subacute and chronic phases (Figure 6F). Furthermore, the highest IgA production was found in the PCCnh group in the subacute phase (0.572 ± 0.037 OD), and a significant difference was observed when compared to the nPCC group. Despite the higher production in these groups, most samples were IgA negative, and only five samples (*n* = 31) were IgA positive in PCCnh patients in the chronic phase. Similar results were observed in chronic phase PCCh patients, where only eight samples (*n* = 37) were positive (Figure 6E).

## 4. Discussion

In this work, we sought to disclose the frequency of symptoms and serological analysis of volunteers with PCC in order to gain a better understanding of the host humoral immunoglobulin response and understand the main risk factors or molecular immunological signatures for the development of subacute and chronic PCC.

Differences in health outcomes between men and women have been described in the literature, with sex being considered as having an important influence on the determinants and consequences of health and illness. Our study showed that more women sought medical attention and reported SARS-CoV-2 sequelae, indicating that women were more affected by the disease than men. However, we should consider that in Brazil, women tend to seek medical care more than men, and despite the difference observed in PCC-related symptoms, we cannot affirm that more women are affected by PCC [22,23].

The most common symptoms observed in this research included fatigue, myalgia, memory loss, hypoprosexia, headache, anxiety, ageusia, anosmia, and reasoning difficulties, many of which involve wide organ or system coverage and are not restricted to only one organ or system, demonstrating how complex and multifaceted PCC is [24,25]. Studies conducted in different regions of the world, such as the United States [9] and Europe [7], obtained similar results to ours in terms of symptom frequency, with high racial diversity (miscegenation) being observed.

When we analyzed symptoms and considered the subacute and chronic phases of PCC, the frequency of symptoms was higher in the subacute group. The progression of symptoms over time revealed three distinct patterns that provide insight into the etiologies and mechanisms underlying COVID-19 sequelae. First, we observed a decrease in the prevalence of symptoms such as ageusia and anosmia from 60% to 30% up to 3 months after symptom onset, remaining at 30% after 12 weeks. This evolution of symptoms over the months indicates that there was a slower recovery from the acute phase, with a number of symptoms remaining in the subacute phase. Second, for other symptoms, we showed that their prevalence increased over time until the chronic phase, such as memory loss and hypoprosexia, which increased from 38% and 39% to 61% and 58%, respectively. Third, some symptoms, such as fatigue and myalgia, do not change in prevalence over time. Interestingly, late onset of symptoms, especially memory loss, appears after 30 days of SARS-CoV-2 infection and has been associated with long COVID [26,27,28]. Some researchers have studied cognitive symptoms and observed that they are common in PCC. It is well known that viral infections are often associated with excessive activation of inflammatory and immune responses, which in turn can trigger and/or accelerate brain neurodegeneration. Recent findings show that the spike protein from SARS-CoV-2 persists in the plasma of long-term COVID patients for up to 12 months after diagnosis, increasing the probability of it reaching the brain and inducing a neuroinflammatory response [29,30].

In addition, the incidence of PCC symptoms could be a response to a single factor or could even be due to a subset of factors such as hospitalization, vaccination, comorbidities, immunological characteristics, or even the type of SARS-CoV-2 variant.

Since the first case of COVID-19 in Brazil, several viral variants have circulated in the population and have been designated by the WHO based on their assessed potential to be more aggressive, highly transmissible, vaccine-resistant, and their ability to cause more severe disease (hospitalization) or death. Indeed, the alpha and delta variants of SARS-CoV-2 induced high mortality, and the omicron variant, since its emergence in late 2021, has been associated with a more benign outcome, with no hospitalization and lower odds of developing long-COVID sequelae compared to earlier variants [31]. However, our study presents a different scenario, where we found a higher prevalence of PCC (42%) in the year 2022, where the dominant variant in the state of Rio de Janeiro was omicron [32], indicating that not only virulence should be considered when analyzing the risk of developing PCC, but also vaccination status and immunological response.

Some studies suggest that vaccination prior to SARS-CoV-2 infection may reduce the risk of PCC [33,34]. Our results are consistent with current research in indicating that, at the time of infection, most of the volunteers with PCC were unvaccinated, and these individuals were more susceptible to developing PCC, supporting the hypothesis that vaccination is the main preventive strategy for reducing long-term symptoms.

SARS-CoV-2 infection induces cellular and humoral responses to eliminate the virus from the host. Many recent papers have addressed the dynamics of immunoglobulin production during or after infection. However, they do not consider or associate immunoglobulin production with the development of PCC. The present work showed IgG and IgA positive patients in the post-acute phase of COVID-19, and, interestingly, high levels of IgA were found in PCC patients.

IgA is produced by B lymphocytes with T-independent and T-dependent mechanisms in defense against different types of pathogens, such as SARS-CoV-2. It protects epithelial barriers from viruses and modulates immune and inflammatory responses [35]. In addition, IgA has recently emerged as an inducer of active immunity by controlling the production of cytokines such as IL-6, IL-23, IL-1β, and TNF by human myeloid immune cells, which is a critical step in controlling local and systemic immunity [35]. Previous studies have reported high IgA production in the acute phase of COVID-19 and a marked decrease in IgA in the following months [36,37]. Because of this, drug therapies have been developed to modulate and enhance IgA production to treat SARS-CoV-2 infection [38,39]. On the other hand, we have previously observed IgA production for a prolonged period of at least 2 months after infection with SARS-CoV-2, but no correlation with PCC was investigated in this case [20]. Cruz et al. (2023) [40] reported higher levels of IgA against the N and S viral proteins in PCC patients, especially in those with severe acute disease. Now, in this work, we observed that PCC patients were positive for IgA, indicating that this immunological factor could be a key feature for understanding the mechanisms underlying PCC.

However, unlike Cruz et al. (2023) [40], the present work showed a different scenario in which IgA production was not only observed in severely acute patients. Hospitalized PCC patients were IgA positive, although non-hospitalized patients were also positive, indicating that hospitalization or disease severity was not the main factor influencing IgA production in PCC patients. The physiological causes of IgA positivity require further investigation, but possibilities include exacerbated inflammation, genetic predisposition, dysregulation of system homeostasis, and age [35,40,41,42,43].

Previous work by our group has shown that older people produce high levels of IgA during SARS-CoV-2 infection compared with young adults and adolescents [20]. The immunological basis underlying the age groups and COVID-19 is still not clear. Adolescents and young adults have attenuated immune responses that culminate in more tolerance to the virus. Our findings revealed that adults and elderly PCC patients produced more IgA than young adults and adolescents. On the other hand, adolescents and young adults may have a more active innate immune response that culminates in efficient virus elimination and low antibody levels [44,45,46] and, consequently, low IgA levels when PCC develops. In addition, the expression of angiotensin-converting enzyme II (ACE-2), the main receptor of the SARS-CoV-2 virus, varies according to age, with young people having high expression of ACE-2 compared with adults and seniors [47]. Such high expression of ACE-2 appears to exert anti-inflammatory properties and could explain the lower IgA production observed in this work with young adults.

Finally, no specific immunological markers for detecting PCC are available to help the medical community in the diagnosis, treatment, or prevention of long COVID development. The results of the present study indicate that the detection of IgA antibodies against SARS-CoV-2 during the course of the disease could be associated with the development of PCC and may be an immunological signature to predict the prolonged symptoms and/or sequelae in COVID-19 patients.

## Figures and Tables

**Figure 1 viruses-15-01545-f001:**
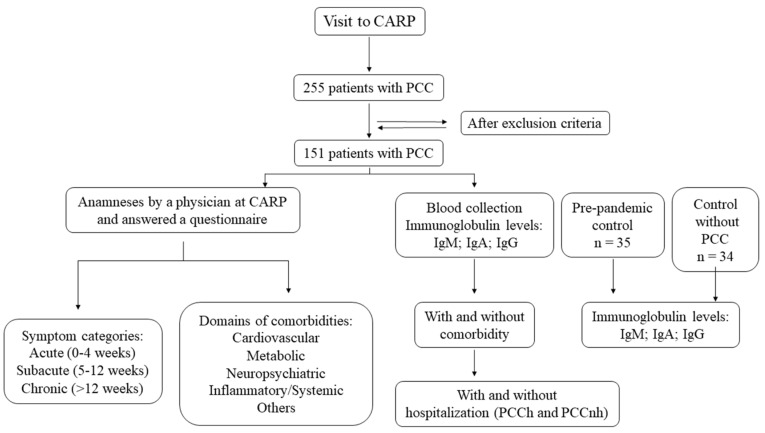
Flowchart of the post-COVID-19 condition (PCC) patients who sought care at the Centro de Acolhimento e Reabilitação Pós-COVID-19 (CARP).

**Figure 2 viruses-15-01545-f002:**
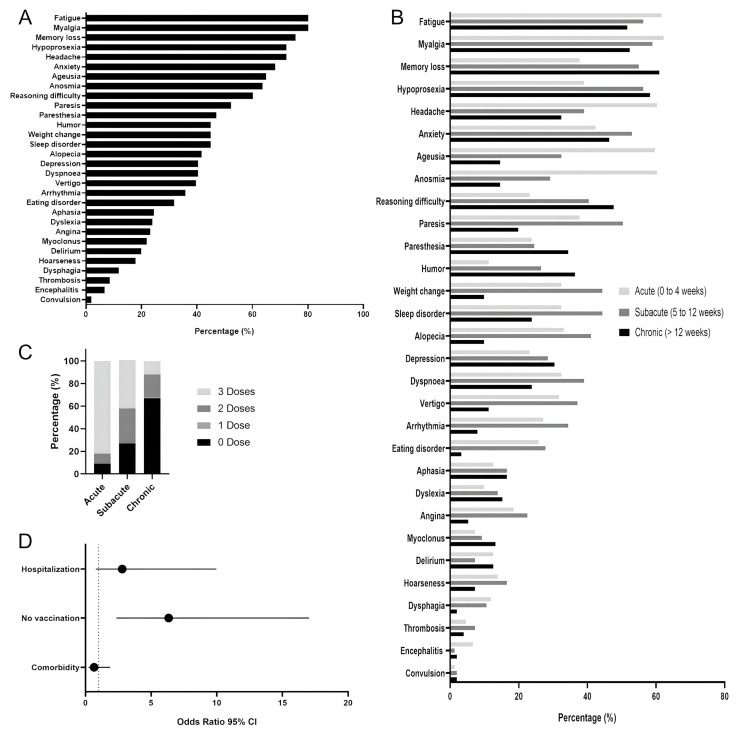
Overview of 151 patients with post-COVID-19 conditions (PCC). (**A**) Frequency (%) of post-COVID-19 symptoms. (**B**) Frequency (%) of symptoms reported in the acute, subacute, and chronic phases of PCC. (**C**) Vaccination schedule of subacute and chronic patients at the moment of infection. (**D**) Odds ratio with 95% confidence interval (CI) analysis of predisposition to the development of chronic PCC symptoms.

**Figure 3 viruses-15-01545-f003:**
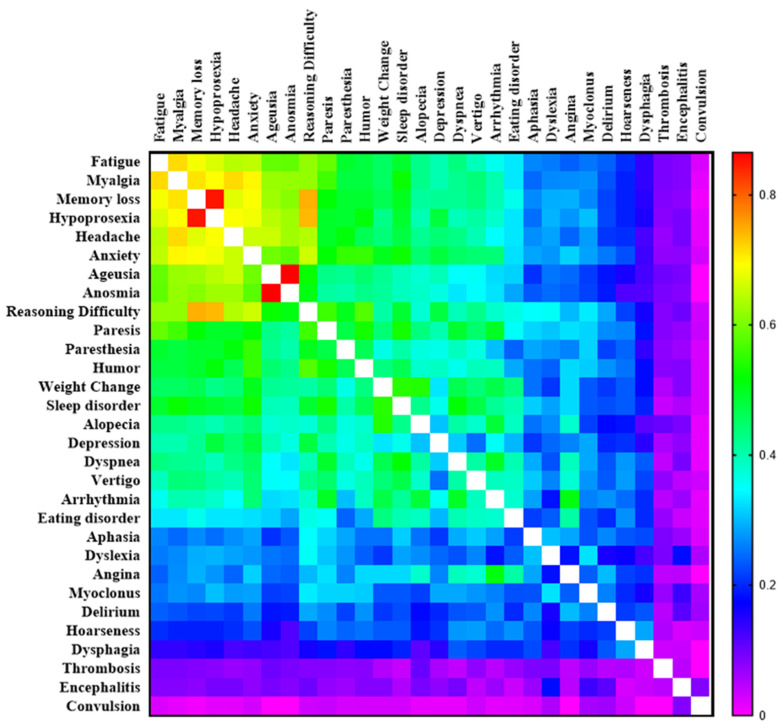
Heatmap showing co-occurrence levels (Jaccard similarity index) between pairs of post-COVID-19 condition symptoms. The Jaccard index was measured based on the presence/absence of symptoms for each patient. Increasing similarities are indicated by a warm color. Values equal to 1 are indicated by a blank square.

**Figure 4 viruses-15-01545-f004:**
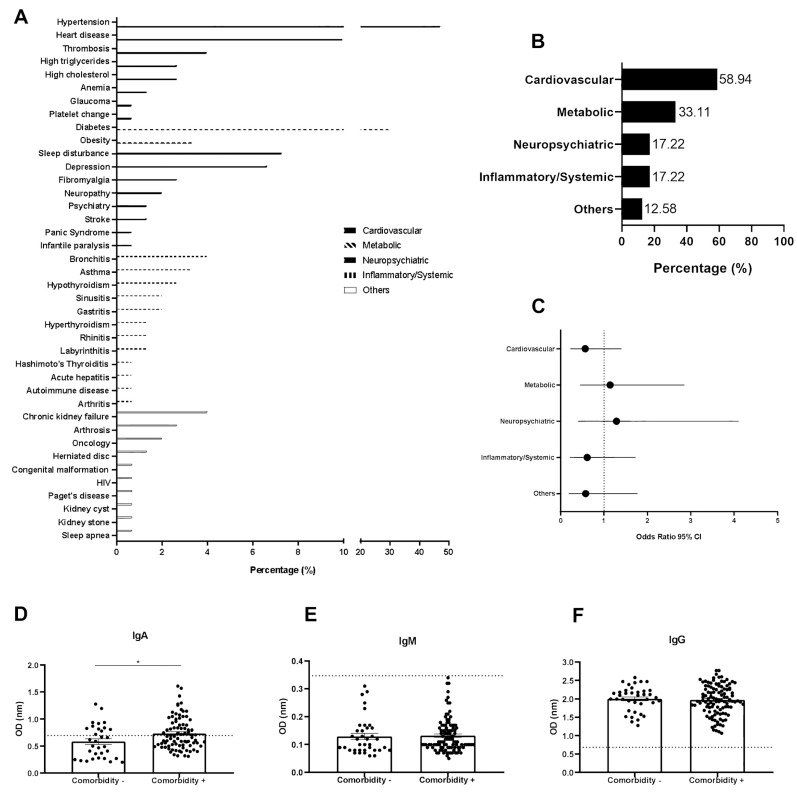
Comorbidities as predisposing factors for the development of post-COVID-19 conditions (PCC). (**A**) Profile and frequency (%) of comorbidities. (**B**) Frequency (%) of comorbidity domains. (**C**) Odds ratio with 95% confidence interval (CI) analysis of comorbidity domains for the development of chronic PCC symptoms. Profile of IgM (**D**), IgA (**E**), and IgG (**F**) comparing the presence or absence of comorbidities. * Significant statistical difference by Mann–Whitney test (*p* < 0.05).

**Figure 5 viruses-15-01545-f005:**
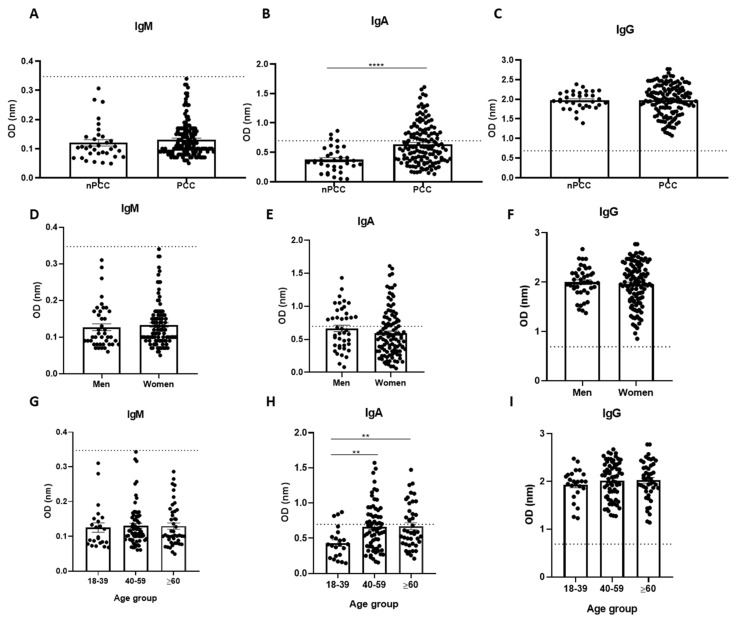
Profile of IgM, IgA, and IgG production in post-COVID-19 condition (PCC) patients. (**A**–**C**) Comparison of immunoglobulin production between patients without PCC (nPCC) and those with PCC. (**D**–**F**) Comparison of immunoglobulin production between men and women with PCC. (**G**–**I**) Comparison of immunoglobulin production between age groups (years). Significant statistical difference using the Kruskall–Wallis test for the analysis of more than two groups followed by Dunn’s post hoc test and using the Mann–Whitney test for two groups. The symbols indicating statistical significance are ** *p* < 0.01; **** *p* < 0.0001.

**Figure 6 viruses-15-01545-f006:**
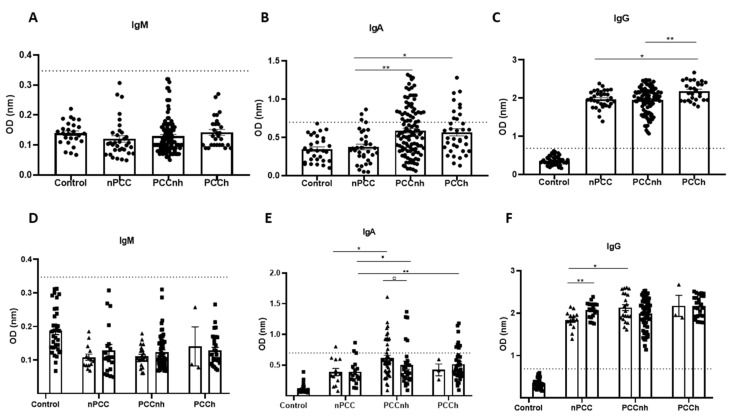
Profile of IgM, IgA, and IgG production of hospitalized and non-hospitalized post-COVID-19 condition (PCC) patients. (**A**–**C**) Comparison of IgM, IgA, and IgG production between the control group (SARS-CoV-2 unexposed), those without PCC (nPCC), those with PCC and non-hospitalized (PCCnh), and those with PCC and hospitalized (PCCn). (**D**–**F**) Comparison of IgM, IgA, and IgG production between the control, nPCC, PCCnh, and PCCn patients divided into subacute (▲) and chronic groups (■). Significant statistical difference using the Kruskall–Wallis test for the analysis of more than two groups followed by Dunn’s post hoc test and using the Mann–Whitney test for two groups. The symbols indicating statistical significance are *,▪,▫ *p* < 0.05; ** *p* < 0.01.

## Data Availability

The data are contained within the article.

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
