# Peer review of "Immunoglobulin A as a Key Immunological Molecular Signature of Post-COVID-19 Conditions"

_viruses, 2023, doi:10.3390/v15071545_

Round 1
Reviewer 1 Report
the paper investigates the pathogenetic role of IgA levels in the pathogenesis of the post covid condition. the paper appears to be interesting, however it requires some modifications before publication
in introduction it is not true that the covid became a pandemic at the end of 2020, the pandemic started from the end of the first half of 2020.
the objectives of the study do not according to the title (…describe main symptoms etc… . vs IgA as a key immunological etc).
if IgA is the target analyzed, correlation between symptoms and comorbidity variables and IgA levels should be made.
the discussion must therefore be written highlighting the data emerging from the new analysis. in the discussion it would be useful to speculate on the role of covid drugs that act on IgA (such as pidotimod) (see doi 10.3390/microorganisms10112131., 10.3390/jcm10245765., 10.4084/MJHID.2020.048.)
Author Response
We thank the reviewers for the comments, all the changes in the article are in red.

Reviewer 2 Report
Please provide data with numbers in supplementary tables, including SD, n, mean and statistics.
Provide a table of patient and volunteer demographics, e.g. age, BMI, sex, comorbidities, with mean and SD.
Methods: What type of blood samples were taken? EDTA?
Confirm in the methods section, that OD values were only taken in the linear range of the assay. Best, show an example of the control dilution.
Figure 2B is illegible and should be similar to Figure 1A.
Please state the number of patients analysed in the legend to Fig 2.
Please always clarify, when total IgG or specific anti SARS-CoV-2 IgG was analysed.
Author Response

(The authors gave the same response as above.)

Reviewer 3 Report
I am not convinced that measuring immunoglobulin levels tells you antyhing about the goal defined by the authors: "in order to gain a better understanding of the host immune response"
r 394.
what do the authors mean with this part:
Authors should discuss the results and how they can be interpreted from the per- 394 spective of previous studies and of the working hypotheses. The findings and their impli- 395 cations should be discussed in the broadest context possible. Future research directions 396 may also be highlighted.
did the authors also look in to RNA levels or prolonged viremia?
several typo's
I think parts of the template from viruses are still throughout the text
could benefit from editing by a native speaker. In each part of the manuscript sentences are too long.
Author Response

(The authors gave the same response as above.)

Round 2
Reviewer 1 Report
the authors improved the paper , following the indications provided
Reviewer 2 Report
Please, correct the legend of Fig. 2: "patients 151" into "151 patients"
Thank you for the corrections.